# Resemblance of the Recurrence Patterns in Primary Systemic, Primary Surgery and Secondary Oncoplastic Surgery

Davut Dayan [1,†] , Kristina Ernst [1,†], Bahriye Aktas [2] , Raffaela Baierl [3], Susanne Briest [2], Martin Dengler [3], Daniela Dieterle [4], Amelie Endres [5], Kathrin Engelken [6], Andree Faridi [7], Hannes Frenz [5], Peer Hantschmann [8], Wolfgang Janni [1], Christina Kaiser [7], Thorsten Kokott [6], Stefanie Laufhütte [8], Florian Schober [9] and Florian Ebner [10,*]

1　Frauenklinik, Universität Ulm, 89075 Ulm, Germany
2　Universitätsklinik Leipzig, Frauenklinik, 04103 Leipzig, Germany
3　Brustkrebszentrum Passau, Klinikum Passau, 94032 Passau, Germany
4　Brustzentrum Kaufbeuren, Klinikum Kaufbeuren, 87600 Kaufbeuren, Germany
5　Medical Facility, Universität Tübingen, 72016 Tübingen, Germany
6　Brustzentrum Stade, 21682 Stade, Germany
7　Brustzentrum, Universität Bonn, 53127 Bonn, Germany
8　Klinikum Altötting, Frauenklinik, 84503 Altötting, Germany
9　Plastische Chirurgie, Diakoneo Schwäbisch Hall, 74523 Schwäbisch Hall, Germany
10　Frauenarztpraxis Freising, Marienplatz, 585354 Freising, Germany
*　Correspondence: ebner@gyn-freising.de
†　These authors contributed equally to this work.

**Abstract:** Purpose: Surgical interventions tend to have an effect on the generation of recurrences in tumor patients due to the anesthesia involved as well as tissue damage and subsequent inflammation. This can also be found in patients with breast cancer. Methods: In this multicenter study, we investigated data of 632 patients with breast cancer and the subsequent diagnosis of a recurrence. The patient data were acquired from 1 January 2006 to 31 December 2019 in eight different centers in Germany. The data sets were separated into those with primary surgery, primary systemic therapy with subsequent surgery, and reconstructive surgery. Three different starting points for observation were defined: the date of diagnosis, the date of first surgery, and the date of reconstructive surgery, if applicable. The observational period was divided into steps of six months and maxima of recurrences were compared. Furthermore, the variance was calculated using the difference of the distribution in percent. Results: The descriptive analysis showed no resemblance between the groups. The variance of the difference of the recurrence rates analysis using the surgical date as the starting point showed similarities in the age subgroup. Conclusion: Our clinical analysis shows different metastatic behavior in different analysis and treatment regimes. These findings justify further investigations on a larger database. These results may possibly identify an improved follow-up setting depending on tumor stage, biology, treatment, and patient factors (i.e., age, . . . ).

**Keywords:** breast; cancer; recurrence; pattern; oncoplastic

## 1. Background

The therapeutic options for breast cancer range from surgery over radiotherapy to systemic treatment agents. Recently, primary systemic treatment (PST) has become the standard in most tumor biology sub-types and stages [1–4]. PST is recommended by multiple guidelines and leads to a 'delay' of the surgery up to 20 weeks. Overall, PST is associated with a higher rate of breast-conserving treatments in operable breast cancers [5].

With systemic treatment, circulating inactive tumor cells are increasingly successfully targeted in breast cancer. This is one of the reasons why breast cancer is considered a systemic disease. In the follow-up, inactive tumor cells may lead to recurrences even after a

long time [6–8]. In this regard, numerous factors have been identified, which may activate or suppress tumor growth [9]. Analyzing the recurrence pattern of their patients, Demicheli et al. as well as Dillekas et al. have expressed concerns in their respective studies that surgical intervention may be one activating factor in this regard [6,10,11]. Dillekas et al. investigated the recurrence pattern after primary surgery and after subsequent reconstructive surgery and found a resemblance of both patterns. They concluded that the underlying mechanism depends on the biochemical cascades following surgery in general [6]. These cascades had already been postulated earlier when Hashimoto et al. investigated patients operated for lung carcinoma. They found a significantly increased number of circulating tumor cells, concluding that tissue destruction was a major factor for activation [12]. Additionally, Ananth et al. suggested in their study that the signaling cascades and the changes of the microenvironment may be triggered by surgical interventions [13]. In regard to breast surgery, Retsky et al. found a correlation of the surgical removal of the tumor and the occurrence of early recurrence with respect to the elapsed time after surgery [8]. Sennerstam et al. went even as far as saying that even core needle biopsy of a breast tumor may change the course of the disease. They found a significantly increased rate of distant tumor metastasis after biopsy, which correlated with the diameter of the needle involved [14]. Hobsen et al. had already found earlier in a mouse model that after the biopsy of a tumor an increase of neutrophilic granulocytes as well as tumor-cells would be observed, suggesting an association of inflammation in the microenvironment with the occurrence of recurrences [15]. Many other authors have described this phenomenon [16–18], while others were not able to verify these results [19,20].

The basic principle behind the correlation of immunologic, microenvironmental changes, and enhanced tumor growth is intelligible if the interaction of the tumor growth and its immunological response is taken into account. This response generally consists of three phases: the "Elimination-phase" (identification and destruction of tumor cells), the "Equilibrium-phase" (tumor cells are not identified; however, further growth is suppressed), and the "Escape-phase" (tumor cells escape the regulatory influence of the immune system and tumor growth is uninhibited). Some authors postulate that a surgical intervention may help to induce this last phase [21].

Based on Retsky and Dillekas' results, our study question is:

Is there a resemblance in the recurrence pattern after surgery in patients with PST, primary surgery, and oncoplastic surgery at a later stage in the follow-up period?

## 2. Methods

To support our hypothesis, a retrospective, multi-center study was initiated and compared data from breast cancer patients with respect to their primary therapy or reconstructive surgery and disease-free survival. All patients with primary diagnosis after 1 January 2006 and recurrence within the first 5 years or oncoplastic surgery and recurrence within 5 years were included. Only complete data sets were collected. By using the database of certified breast centers, it was ensured that the primary therapy was conducted according to the current guidelines. The main inclusion criterion was a diagnosed recurrence within four years of the surgery. R1-resections were excluded as well as all patients with other malign diseases or primary metastatic breast cancer.

All patients included in the study were classified into three groups with respect to the time of surgery within the treatment regime. A flow chart of the three groups is provided in Figure 1. Group 1 consisted of all patients who received a primary surgery with an adjuvant therapy and later developed a recurrence without further recorded surgical intervention in the meantime. Group 2 consisted of those patients who received a PST with delayed surgery who again later developed a recurrence without further recorded surgical intervention in the meantime. PST did not include anti hormonal treatment prior to surgery. Group 3 consisted of those patients who were treated according to the treatment regime of group 1 or 2 and later received a secondary reconstructive surgery before developing a local or distant recurrence.

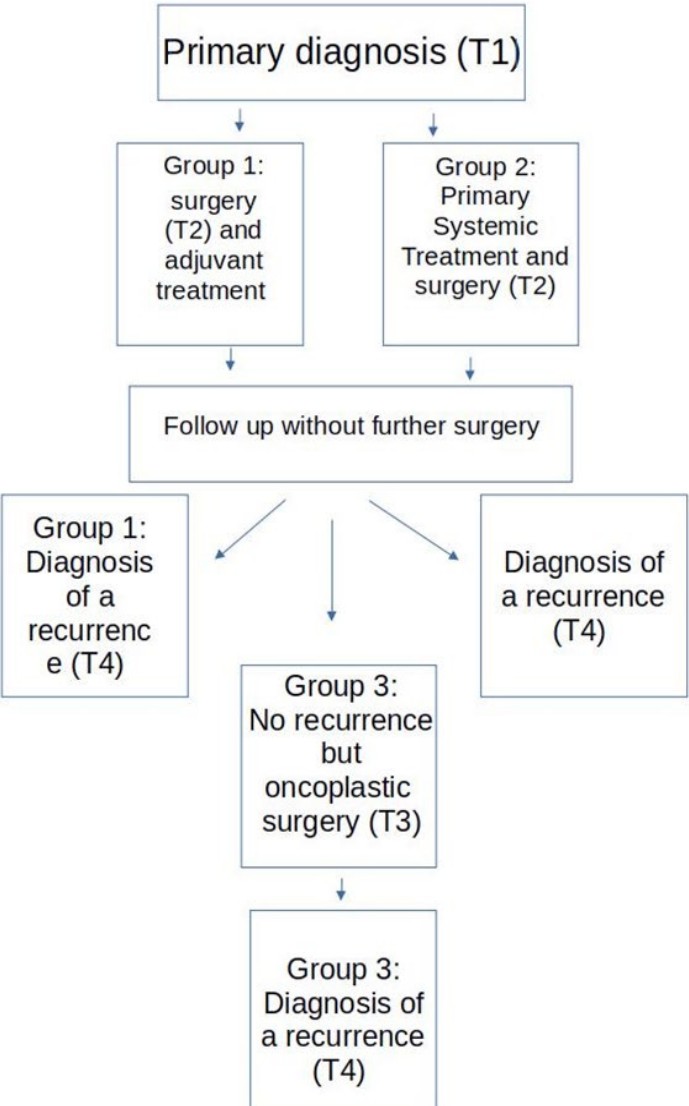

**Figure 1.** Study flow chart.

Data were assessed in two university and six peripheral certified breast centers in Germany. As certification was a requirement, all centers were associated with a tumor-register and a department for plastic surgery. Data was collected over 18 months and was stopped on 1 February 2020. Data were transmitted and anonymised electronically according to the European data safety requirements. Variables collected included patient data, such as age and tumor stage, information on therapeutic approaches, and date of recurrence (T4). T4 could be the date of biopsy in the case of regional recurrences or the date of CT, MRI, or scintigraphy in the case of distant recurrences. The diagnosis of a recurrence of breast cancer or a finding that was highly suspicious for such a diagnosis within 5 years was an inclusion criterion.

For the recurrence-free survival, three time periods were relevant: date of diagnosis (T1) until T4 (t1), date of surgery (T2) until T4 (t2), and date of reconstructive surgery (T3) until T4 (t3).

A descriptive analysis as well as a risk assessment were conducted. The time interval t1 was considered as the dependent variable of our study. For the analysis of the recurrence pattern, the three recurrence-free survival measures mentioned above were summed for each group for a certain point of time and the square root was taken of this sum. The underlying formula used was $d = \sqrt[2]{\sum_{i=1}^{n} D^2}$ with $d$ as the time interval to the recurrence and $D$ the individual patients time interval to the recurrence. The recurrence-free survival

was grouped in steps of six months, so called SMHR (six months hazard rate). For these steps, the relative rate of recurrence for each group was calculated and plotted against the recurrence-free survival. Maxima was defined as higher values compared to the neighboring values on each side. Subsequently, the recurrence patterns were compared with one another depending on the beginning of observation. As by inclusion criteria, group 1 and 2 were not allowed a reconstructive date (T3); for these groups, t2 was used to compare with t3 of group 3. A comparison was made for the different maxima of the recurrence rates.

A Kaplan–Meier analysis was performed to show any significant difference between the recurrence rates of the groups. In order to better quantify the differences over time, a direct comparison of the curves is needed. Due to the different case numbers in each group, the relative recurrence rate was calculated for further analysis. The relative recurrence rates in percent were subtracted from the baseline recurrence rate (group 1). Of the resulting differences, the variance was calculated providing a better estimation.

This study protocol was reviewed and approved by the ethic commission of the university of Ulm, approval number 294/18.

## 3. Results

After all inclusion criteria were met, 481 patients of eight clinics met the inclusion criteria of which 390 (81.1%) were in group 1, 72 (15.0%) in group 2, and 19 (4.0%) in group 3. In our study, PST delayed the definite surgical treatment on average for 4.7 months between group 1 and 2 (t2: group 1: 1 month, group 2: 5.7 months). Within the groups, there was a different distribution of intrinsic sub-types due to the indications for PST and adjuvant treatment (Table 1).

**Table 1.** Patient and tumor characteristics, Δ = time difference, TNBC = triple negative breast cancer, age when diagnosed in years.

| | | Groups *n* (%) | | |
|---|---|---|---|---|
| | | **1 (*n* = 390)** | **2 (*n* = 72)** | **3 (*n* = 18)** |
| Intrinsic Tumorsubtype | Luminal A | 156 (41%) | 17 (24%) | 7 (39%) |
| | Lum. B HER2-neg. | 28 (7%) | 14 (19%) | 0 |
| | Lum. B HER-2-pos. | 63 (16%) | 10 (14%) | 0 |
| | HER2-pos. | 69 (18%) | 10 (14%) | 2 (11%) |
| | TNBC | 36 (9%) | 15 (21%) | 3 (17%) |
| | ? | 38 (10%) | 6 (8%) | 6 (33%) |
| Tumordiameter | T = 1 | 189 (48%) | 30 (42%) | 11 (61%) |
| | T > 1 | 201 (52%) | 42 (58%) | 7 (39%) |
| Status of lymphatic nodes | N0 | 155 (40%) | 24 (33%) | 11 (61%) |
| | N+ | 219 (60%) | 46 (64%) | 7 (39%) |
| Usage of ionizing radiation | | 280 (72%) | 52 (72%) | 9 (50%) |
| Age when diagnosed | ≤50 | 110 (28%) | 42 (58%) | 6 (33%) |
| | >50 | 280 (72%) | 30 (42%) | 12 (64%) |
| Location of recurrence | Local | 223 (62%) | 42 (58%) | 12 (67%) |
| | Distant | 156 (11 both) | 27 (3 both) | 6 (2 both) |
| ΔT (diagnosis and surgery) | In months | 1.05 | 5.8 | |

The recurrences over time for the groups and subgroups are provided graphically in addendum 1 (Supplementary Materials). Looking at the t1 results of the recurrence graphs, group 1 has maxima at 18, 30, 48, and 60 months follow-up. Group 2, on the other hand, has maxima at 12, 24, and 42 months. So, these maxima appear 6 months earlier compared to group 1. Starting the follow-up at the surgery date (t2—Figure 2), the peaks in the graphs were less prominent. In group 1, the first two maxima do not change, but the

42- and 60-month peaks become less prominent. In group 2, the first maximum appears 6 months earlier as expected, but the 24 maximum levels out and the 42-month maximum does not change. Group 3 shows maxima at 12, 24, and 36 months follow-up.

These were for group 1 at 18, 30, and 60 months with a plateau at 6 and 42 months. The interval between the first two peaks was 12 months. Group 2 showed a maximum at 6 and a second, smaller peak at 42 months. For group 3, the maxima were at 12, 24, and 36 months, also with intervals of 12 months. Next, a Kaplan–Meier survival analysis was performed The *p*-value of the comparison equals 0.00853280, $(p(x \leq \chi^2) = 0.991467)$ showing no difference in the overall recurrence rates.

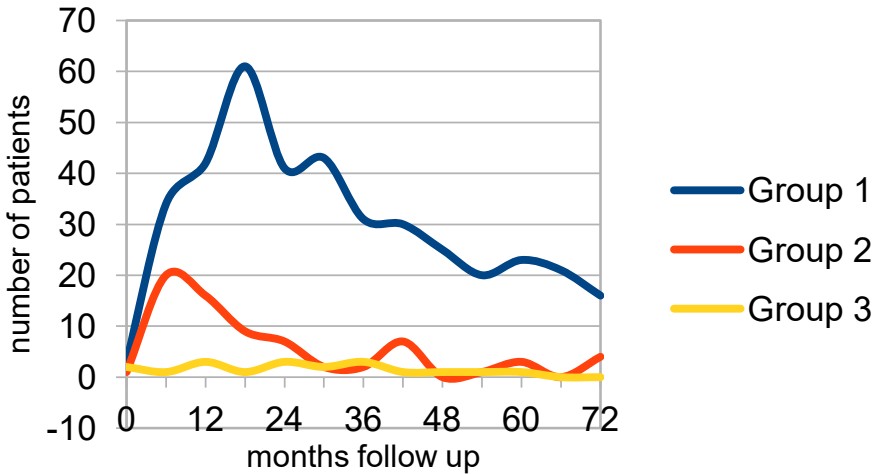

**Figure 2.** Surgery to recurrence (t2/t3) group 1 (*n* = 390), group 2 (*n* = 72), group 3 (*n* = 16).

Subgroup analysis for intrinsic sub-types, nodal status, radiotherapy, and age was performed for group 1 and 2.

As different subgroups have a different metastatic pattern, the next analysis visualized the recurrences of the different intrinsic sub-types. The largest group was the luminal A tumor patients (*n* = 173). In the t2 graphs, the maxima were more prominent in both groups. In group 1 t1, the first maximum was at 24 months, but in t2 a 6-month peak appeared. In group 2 t1, there was double peak at 6 and 24 months, similar to the peaks in group 1. However, in the t2 graph, a plateau from 6 to 15 months showed.

In TNBC (*n* = 51), a first peak at 6 months was seen in both t2 groups, with a correlating peak in group 2 t1 at 12 months. The pattern in group 1 t1 showed more of a plateau from 12 to 30 months.

In Her2+ (*n* = 79), group 2 had a peak at 6 (t2) and 12 months (t1), while group 1 peaked at 18 months (t1) and had a peak at 12 months (t2). Contrary to luminal A and TNBC, the Her2+ subgroup had a low at 24 months f/u.

The luminal B Her2+ subgroup (*n* = 73) had nearly identical graphs in group 1 up to 54 months f/u. The peaks were at 6, 18, and 30 months. In group 2, the time shift of the first peak from 12 to 6 months is visible.

The luminal B Her2- subgroup (*n* = 42) showed a time shift for the first peak in both groups. In both groups, the first peak appeared at 18 months (t1) and 12 months (t2). The second peak in group 2 also shifted from 30 months (t1) to 24 months (t2). However, in group 1, the second peak at 30 months (t1) shifted to 36 months (t2), prolonging the interval between the two peaks from 12 months to 24 months.

The age subgroups shown in group 1 had peaks at 6 months for the younger patients (t1) and the older patients (t2). In those two groups, a second peak is at 18 months, whilst in the other two groups, the recurrence peak is at 12–18 months (younger t2) and 18 months (older t1). There is a low at 42 months (older) and 36 months (younger patients) in both timelines.

The nodal status as a surrogate for a more advanced tumor state was our next analysis. Nodal positive (pN+) and nodal negative (pN−) patients in each group were compared.

The results of the nodal status sub-analysis showed for group 1 pN− and pN+ a time difference of 12 months for the first peak (pN−/pN+: 6/18 months), with the pN+ patients having earlier recurrences. Similar in group 2, there was pN−/pN+: 6/12 in the t1 distribution and a time difference of 6 months. The pN− distribution had a peak at 12 and 30 months follow-up in group 1 and a 12 and 24 months (t1) versus a 6 month (t2) follow-up. Comparing the pN+ subgroup, a 6 and 18 month peak in group 1 and in group 2 a 6 (t1 and t2) and 24 months peak (t1) with a slower decline at 18 months (t2) is identifiable. For pN+, in group 1 the time difference between the peaks was 12 months and in group 2 18 months. The inter group comparison showed a 6-month peak in all pN+ t1 patients and group2 pN− t2. The other pN− subgroups showed a peak at 12 months. If a second peak appeared, it was 12 or 18 (group 2 pN+) months after the first one.

So, in the PST group, the pN+ subgroup had no change in the recurrence pattern (peak at 6 months), whilst in the pN0 group, the expected delay due to the PST can be found. The pN+ patients had the first recurrence maximums at 6 months f/u regardless of the treatment regime (PST or adjuvant).

Radiotherapy as a treatment modality with known reduction of local recurrences was also analyzed for the recurrence pattern. In group 1, four peaks at the same time are identifiable. In group 2, these were not as clearly defined except for the t2 group with radiotherapy. The first peak at 6 months was the subgroup of treated patients in group 1 contrary to group 2 where the 6 months peak appeared in the untreated patients. The peaks of the recurrences did remain at 6 or 12 months irrespective of the starting date (t1 or t2).

Next, the difference between two (sub)groups of the relative recurrence rate per time interval was calculated and the variance of these numbers was used as a measure of the variability (Table 2). As internal control, the t1 vs. t2 of group 1 and group 2 was used. Here, a variance for group 1 (t1 vs. t2) of 1.6 and for group 2 (t1 vs. t2) of 30.2 was found, showing a good correlation for group 1 and indicating the expected shift in recurrences in group 2.

**Table 2.** Variability of the relative recurrence rates in the direct subgroup comparison. Intern: comparing the t1 vs. t2/t3, y = years, LumA = luminal A, Bher− = luminal B Her2 negative, Bher+ = luminal B Her2 positive, TNBC = triple negative breast cancer, Her3 = Her2 overexpression.

| Varianz | Intern<br>t1 vs. t2 | Group 1<br>t2 vs. t2/t3 | | t1 vs. t1 | Group 2<br>t2 vs. t3 |
|---|---|---|---|---|---|
| group 1 | 1.6 | - | - | - | - |
| group 2 | 30.2 | 15.9 | - | 57.8 | - |
| group 3 | - | 33.6 | - | - | 30.5 |
| age | <50 y vs. ≥50 y | <50 y | | ≥50 y | |
| group 1 | 8.5 | - | - | | |
| group 2 | 32.5 | 32.5 | - | 95.2 | |
| group 3 | - | - | - | - | |
| Nodal status | pN0 vs. pN+ | pN0 | | pN+ | |
| group 1 | 20.7 | - | - | - | |
| group 2 | 13.1 | 84.8 | - | 41.4 | |
| group 3 | - | - | - | - | |

| Subtypen | intern | | | | | group 1 | | | | |
| | LumA | Bher− | Bher+ | TNBC | HER+ | LumA | Bher− | Bher+ | TNBC | HER+ |
|---|---|---|---|---|---|---|---|---|---|---|
| group 1 | 3.4 | 17.6 | 5.2 | 18 | 6.1 | - | - | - | - | - |
| group 2 | 30 | 125.1 | 16.7 | 132.2 | 100 | 52.4 | 88.7 | 118.9 | 232.1 | 95.3 |
| group 3 | - | - | - | - | - | - | - | - | - | - |

The substraction of the variance for t2 group 1 vs. group 2 was 15.9, 33.6 for group 1 vs. group 3, and 30.5 for group 2 vs. group 3. For t1 group 1 vs. group 3, the result was 27.2. Subgroup analysis for intrinsic sub-types, nodal status, radiotherapy, and age was performed for group 1 and 2. A substraction of the variance <30 was only found for the subgroups <50 years vs. ≥50 years (group 1: 8.5; group 2: 28.2; <50 group 1 vs. <50 group 2: 31.7). The other subgroups (intrinsic sub-types, pN+, radiotherapy, and T-stage) had larger variances.

## 4. Discussion

Even though it is suspected that the recurrence rate may increase after oncoplastic surgery [6], this observation has to the best of our knowledge not been published in a group comparison of neoadjuvant/adjuvant treated patients and oncoplastic patients with the surgical date as the starting point. In case surgery may stimulate a recurrence, we hypothesized that the recurrence should appear 6 months later in patients with PST. In order to reduce possible center specific factors such as pain/drain management, etc., our study was designed as a multi-center retrospective study.

The intrinsic sub-types and tumor stages in our three study groups reflect the different recommendations for neoadjuvant/adjuvant systemic treatments. Group 2 consisted of more aggressive intrinsic tumor sub-types, higher tumor classification, higher rates of affected lymphatic nodes, and younger average age. Thus, earlier recurrences in group 2 t1 could be expected. The influence of these factors on the risk of recurrences has been investigated in former studies [22–25]. The pathomechanisms behind the recurrence pattern and intrinsic sub-types have not yet had a clinical impact. As this study aims to identify correlations between surgery and recurrence rates, these factors are further investigated.

The general pattern of recurrence in breast cancer has been investigated in other studies. In one study with 2451 patients, this pattern was measured from 12 to 24 months after diagnosis [22]. Confirmed by other studies [6,23,24,26,27], our data shows this peak for the complete group 1, but group 2 (t1) showed the peak at 6 months. Further analysis identified subgroups with an earlier recurrence peak at 6 months (i.e., group 1 with radiotherapy or pN+ or luminal B Her2+, group 2 without radiotherapy or pN+ tumors). This indicates different biological behavior with more unfavorable biology having earlier recurrences. With surgery as the starting point, the recurrence pattern changed. As anticipated for group 2, most of the recurrences were found about 6 months earlier. However, in some analyses this did not occur, such as the radiotherapy analysis. Here, only the height of the peaks changed but the time interval remained contrary to Demicheli et al. [28]. This could be due to the fact that the radiotherapy may remain at the same position in the treatment sequence (i.e., after chemotherapy and surgery) and therefore the possible effects are not evident in our analysis. Another possibility would be that radiotherapy improves the disease-free survival [29–31] by reducing the recurrence risk but cannot change or slow the underlying biological recurrence processes in some patients. The reasons might be the different susceptibility of tumor sub-types to radiotherapy [32] or in a change in microenvironment [33]. As radiotherapy is indicated with breast conserving surgery or an advanced pN+, the earlier recurrences may result from advanced tumor biology (i.e., higher gnomic instability).

Contrary to [22,23], in our results the pN+ subgroup had earlier recurrences in the neoadjuvant treatment regime compared to the adjuvant regime. Only after taking the surgical date as the starting point the first recurrence peak was identical at 6 months.

Younger age is widely considered as an independent risk factor for recurrences [23,34–39]. Interestingly, our analysis shows an early peak in group 1 and group 2 t2 at 6 months. This would support our hypothesis in younger patients. These patients are more likely to have had a PST treatment and unfavorable tumor biology, which indicates different clinical behavior of the different intrinsic sub-types. Interestingly, a similar effect was seen in the older patients with adjuvant treatment. A possible explanation may be the different perioperative handling of older patients [40]. As older patients tend to have

more comorbidities, the effect of intraoperative medication and post-surgical aftercare may contribute to this effect.

Besides higher tumor stages, the initial intrinsic sub-type also defines different metastatic behavior.

Luminal A is commonly considered the most 'benign' sub-type. Our results indicate a more defined time interval from surgery to recurrence in the adjuvant setting, but in the neoadjuvant setting the pattern is less clear. This might be due to the fact that only advanced luminal A tumors are recommended systemic treatment and therefore the numbers in group 2 are limited. Our data confirm the recurrence pattern of Luminal A tumors published by Lim et al. The researchers found a maximum of the recurrence rate two years after diagnosis [25]. Metzger-Filho et al. conducted a similar investigation and observed a maximum after three to four years after diagnosis [41], which correlates with our second peak in the luminal A subgroup. Other studies had similar findings [22,42,43].

A more aggressive sub-type, the TNBC, has been treated with PST for several years now. Here, our results show a more prominent first peak in both subgroups, again indicating that the time interval from surgery to recurrence is closer for more patients. Additionally, the interval to the second peak is shortened in the adjuvant setting whilst in the PST setting it seems to be prolonged.

For luminal B tumors, Metzger-Filho et al. and Park et al. found maxima in the second, fourth, and fifth year after diagnosis. However, in these studies the tumors were not differentiated in luminal B Her2+ or Her2− [41,43].

Other authors did differentiate into further subgroups and found luminal B Her2− tumors maxima in the third, fourth, and fifth year after diagnosis [25,44]. Because no information is given on the data collection, the time intervals are only provided in annual intervals. Our results for the luminal B Her2− subgroup support our thesis in the adjuvant and neoadjuvant analysis. Both groups show an earlier peak in the t2 analysis. Like the adjuvant luminal A group, this could indicate a contribution to the early recurrences by the inflammation following a surgical procedure healing process.

The Her2+ subgroup shows in group 2 the expected shift of 6 months and no changes in pattern. In group 1 again the more prominent peak indicates a clearer defined time interval in t2. This subgroup also shows the expected shift in group 2 but only for the first peak. This may indicate a different metastatic pathway/environment in Her2+ tumors needed for progression compared to the other sub-types [45] and for the early recurrences. Recent research which suggests Her2+ tumors may use inflammatory response pathways [46] to progress could be supported by these clinical findings.

The recurrence pattern of luminal B Her2+ tumors confirmed previous findings [25,41,43,44]. In group 1, the peaks remained at 6 months, whilst in group 2 the peak was earlier and more prominent, again supporting our hypothesis in the PST setting.

Previous publications such as Dillekas et al. or Demicheli et al. [6,10] are confirmed by the results of our group 3 analysis. This supports the theory of a possible underlying pathomechanism. However, as those oncoplastic surgeries are not recorded routinely in the follow-up databases of certified breast cancers, our number for this subgroup is too small and needs to be reconfirmed with a larger database.

Although most weaknesses have been mentioned briefly earlier, we would like to point out the following issues:

The tumor biology as represented in the intrinsic sub-types is not balanced between the groups. As PST becomes more common in most breast cancer sub-types, this will be hard to balance in a retrospective study. On the other hand, our hypothesis was to find a similarity between the recurrence rates starting with the surgical date. As the surgical treatment of the intrinsic sub-types is the same, the risk of recurrence might be higher with more aggressive sub-types but the effect on the time delay is unknown. Our results point to the subgroups of pN+, no radiotherapy, and the luminal sub-types. Again, this would need a larger, more detailed analysis to provide more information.

The small number of patients especially in group 3 limits further in-depth analysis. Unfortunately, neither breast centers nor tumor databases in Germany collect information on oncoplastic surgery. An ideal database would not only note the type of oncoplastic surgery but also other surgeries, as it is unknown whether only local surgeries influence the recurrence risk or if other surgeries such as appendectomy may also increase the risk. As 'previous breast surgeries' have been a very long risk factor for developing breast cancer, in our study the data recording was limited to breast surgeries.

This points to the lack of surgical details in our study. It will be hard to record an unbiased 'surgical trauma score' to estimate the local impact on the microenvironment with current clinical knowledge. This also accounts for the different oncoplastic procedures. In order to have a defined standard, one person collected and classified the surgery according to the theater reports.

As mentioned in the introduction, there is some discussion about anesthesia medication and tumor growth [16,47]. Our study did not collect information on the type of anesthesia, the medication, or duration. This might provide further insights into the tumor pathways or clinical treatment [48,49]. As tumor environment, biology, and clinical outcome are most likely interconnected via multiple pathways, many more variables may be needed to identify the ideal subgroups.

Besides these limitations, our results show an early recurrence peak in most analyzed subgroups. This early peak is delayed in PST but brought forward or made more prominent by using the surgical date as the starting point. These clinical results contribute to the growing understanding of the different tumor sub-types and clinical behavior.

## 5. Conclusions

The study question 'Is there a resemblance in the recurrence pattern after surgery in patients with PST, primary surgery, and oncoplastic surgery at a later stage in the follow up period?' was tested in this multi-center analysis. Comparing patients with upfront surgery and primary systemic treatment, our data supported our hypothesis in the age subgroup.

These clinical findings contribute to the theories and animal studies regarding surgery-related inflammation, immunosuppression, and changes of the microenvironment and their stimulus or facilitation of recurrences. Further studies are needed to confirm our clinical data, understand the interactions, and ultimately investigate the clinical precautions needed to target these pathways.

**Supplementary Materials:** The following supporting information can be downloaded at: https://www.mdpi.com/article/10.3390/curroncol29110698/s1, General analysis of the groups.

**Author Contributions:** D.D. (Davut Dayan), K.E. (Kristina Ernst) and F.E.: idea and initiation of the study, drafting W.J. and B.A. writing of the study protocol and manuscript, B.A., S.B., M.D., D.D. (Daniela Dieterle), K.E. (Kathrin Engelken), R.B., A.F., P.H., C.K., T.K., S.L. and F.S.: Data collection, revision of study proposal and script, A.E. and H.F.: statistical analysis. All authors have read and agreed to the published version of the manuscript.

**Funding:** This research received no external funding.

**Institutional Review Board Statement:** This study protocol was reviewed and approved by the ethic commission of the university of Ulm, approval number 294/18.

**Informed Consent Statement:** Informed consent was obtained from all subjects involved in the study.

**Data Availability Statement:** Data is available upon request with a detailed research question from the corresponding author.

**Conflicts of Interest:** The authors declare no conflict of interest or competing interests.

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
