# Peer review of "Resemblance of the Recurrence Patterns in Primary Systemic, Primary Surgery and Secondary Oncoplastic Surgery"

_curroncol, doi:10.3390/curroncol29110698_

Round 1
Reviewer 1 Report
The authors of the manuscript tested the hypothesis on a large group of patients: 'Is there a resemblance in the recurrence pattern after surgery in patients with PST, primary surgery and oncoplastic surgery at a later stage in the follow up period?'
The results of the presented study are very interesting and even surprising, as they may indicate the lack of influence of preoperative treatment on the time of local recurrence.
I would propose to present the results of local nuns additionally in the table, it would be easier for the reader to interpret.
I would also like to know the authors' opinion whether the results of the study should be translated into the dates of the planned follow-up visits for patients? If relevant, it may be useful to record this suggestion in the manuscript conclusions.
Author Response
Dear reviewer,
thank you for your constructive comments. I assume the ‚nuns‘ should be recurrences which we have added to the table.
Our data is not solid enough to make any recommendations for a change in follow up procedure but there are several ongoing trials that should improve the current evidence behind follow up (Survive study of the German Breast Group).
Florian Ebner
Reviewer 2 Report
The background and topic are good, and only a few significant flaws in the data presentation and discussion. However, logical deduction and demonstration are terrible. And many minor imperfections can be observed, which suggests that the proofreading of the manuscript needs to be done correctly.
1. Font size not consistent in the same paragraph:
Abstract-methods-line4
Abstract-methods-line9-10
Abstract-conclusions
No information needs to be emphasized with the enlarged font size.
2. The abstract section is not conclusive. The information addressed by this section is limited and without logic:
Abstract-results:
1) How about group 3?
2) The second sentence of this section cannot be listed here. It should be more likely to put an abstract-methods section to describe the group definition.
Abstract-conclusion:
1) I don’t understand how this conclusion came up from the results. It seems to be no logical connection there.
2) The conclusion concluded nothing. It only mentioned the limitation of this research on a small sample volume and said that looking forward to furthering studies on the more extensive database. It is without any value.
3. Background section:
Line2-5: What is the definition of PST? Is that surgery with neoadjuvant therapy? What are the differences between PST and other surgery, authors didn’t introduce them.
Subtitle 1.1: The theme of this research is about surgery and recurrence; why here authors didn’t mention it? Why did the authors use micro-environment and tumor growth as critical work here? It seems out of the theme.
Contents of subtitle 1.1: Given that subtitle is defined in this section as about the micro-environment; however, there is no example or discussion about the micro-environment and tumor growth.
Subtitle 1.1-line12-13: What is the authors’ understanding of signaling cascade? Which signaling is involved? Need examples.
Subtitle 1.1-line19-21: In which location was this phenomenon observed? The primary site, recurrent site, or circulation?
Subtitle 1.1-line5-6 from behind: How?
Hypothesis: This is not a hypothesis; it is a question.
4. Method section:
Paragraph 2-line 7-8: Why exclude antihormonal therapy?
Paragraph 3-line 1-2: How can readers access the source data? If the research hopes to publish, the source data with necessary information mentioned in the context should also be attached to the manuscript. Or at least provide the linkage to access the source data. Otherwise, readers will challenge the reliability and authenticity of this research.
Formula: What is the symbol d or D stand for?
Paragraph 5-line1-2: The date of diagnosis of a recurrence cannot be considered as the dependent variable of our study. The Time interval from T1 to T4 should be the dependent variable.
Paragraph 6-line 5: If the effect is synergistic instead of additive, the subtraction here is incorrect. The authors need to exclude this scenario.
5. Results
Paragraphs 6-9: What is the author's standard for the dived subgroup of TNBC, luminal, and Her2+? And can hER2 be negative in luminal B cases?
6. Discussion
Overall, need to be more precise.
7. Conclusion section:
Paragraph 1: The hypothesis should be a description instead of a question.
Paragraph 2: No micro-environment or immune research is included in this work; how can authors conclude with change with micro-environment or immune?
Author Response
Dear reviewer,
thank you for your positive and constructive comments which we are delighted to include in our draft.
En detail these are:
Abstract results: group 3 is now mentioned. The second sentence has been reworded to „The results of the recurrence analysis using the surgical date as starting point showed similarities in (small) subgroups.” to emphasis on the results.
The conclusion has been reworded – as per other reviewers request – to point more towards individualised follow up. ‚Our clinical analysis shows different metastatic behavior in different analysis and treatment regimes. These findings justify further investigations on a larger database. These results may possibly identify an improved follow up setting depending on tumor stage, biology and patient factors (i.e. age, ...). ‘
Primary systemic treatment (PST) is the upfront chemotherapy, also referred to as neoadjuvant chemotherapy (NAC). Recently the upfront endocrine treatment of the tumor has been investigated. Hence our exclusion. As per name primary SYSTEMIC treatment this is not associated with surgery. Please rephrase your question.
The effect of surgery on tissue is not yet fully understood. In order to show the complexity of the issue we try to explain previous published articles contributing to the understanding of the wound healing and known effects. This is summarized in the paragraph microenvironment.
A signaling cascade or pathway describes the interaction and reactions of different agents like prostaglandins or hormones with other agents or cells. As much as we would like to answer the question to which pathway is involved we can not answer this by a simple example as the interactions are multiple and complex. This has not yet – to the best of our knowledge – resulted in a complete understanding of the microenvironment or the surgical effect on it.
We included ‘local or distant recurrence’ in line 19-21 and further in the table the differentiation in distant/local (=primary site) per group. Circulation is currently not monitored routinely – despite the data having at least a 5 year follow up blood monitoring for follow-up was in its very early stages back then. As for Dillekas and Demicheli we included ‘Analyzing the recurrence pattern Demicheli ….’ to explain how these authors generated their hypothesis (line 5-6).
Hypothesis is reworded in ‘study question’
Upfront antihormonal treatment has been excluded due to non availability at the time of primary diagnosis of the patients and ensure this detail is provided as this treatment is currently being discussed as monitoring tool for luminal B tumor behavior.
Data availability is mentioned in Paragraph 5 upon reasonable request.
The formula is defined now by adding ‘with d as the time interval to the recurrence and D the individual patients time interval to the recurrence.’ behind the formula.
Thank you for pointing out the correct dependent variable which is now reworded to ‘The time interval t1 was considered as the dependent variable of our study.’
A synergistic effect is theoretically possible but negative recurrence rates are clinically not possible we believe this scenario does not need ruling out. And the variance is calculated with squared numbers to compensate negative numbers.
A very good question. Our standard for this database is provided in table 1 with a differentiation between HR- Her2+, luminal B Her2+ and luminal B Her2-. We are aware that other authors combine Her2+ regardless of HR-status. Other categories may include BRCA mutation or Her2 low. We had to compromise and find the minimal complete dataset between all breast centers (that is why Ki67 is missing).
With the word limitation and the clinical approach of this draft it is impossible to do the surgical, microenvironment and tumor behavior justice. A precise discussion would also involve animal studies like PMID: 36325601. Again it is a compromise between clinical findings, laboratory results and meta-analysis observations.
However followed your remarks in the conclusion section and reworded the hypothesis to adapt.
Thank you for your constructive remarks which we tried to incooperate into our draft and thank you again for helping to improve our submission
Florian Ebner
Reviewer 3 Report
Dear authors,
Your manuscript is interesting and well-written.
Pay attention because in the abstract you used more than one format and also different dimensions of the letters.
Author Response
Dear reviewer,
thank you for pointing out the formatting errors from docx → pdf file format. We have reformatted the file in one font size and style and payed attention to the proper formating in the pdf file.
Yours sincerely,
Florian Ebner
Reviewer 4 Report
Critique:
In the abstract, the statistical method of difference should be mentioned for clarity. The result should be explanatory in the abstract too and instead of group1 group 2, the variable names should be mentioned.
In the conclusion of the study, it is suggested to use direct relationships discovered instead of stating “certain sub-groups”.
The general background lacks the necessary depth of literature to introduce the pathologies. Please devise a thorough background to the general problem of Breast cancer and its types or subgroups, then treatments and surgery options or radiotherapy thoroughly.
Section 1.1. has not been introduced in 1 and there is no transition from general to specific segments. Surgical procedures or harmful impacts of surgery (inevitable) need to be introduced first.
In the background of the study the specific procedure-related correlation with inflammation or recurrence is not sufficiently described (e.g. different types of interventions and burden of the immunologic response generated/measured).
The methods section needs to be divided into sub-sections for clear understanding such as patient datasets, descriptive analysis, Kaplan maeier analysis, etc.
The differentiation of groups if presented in a tabular form could present a good approach with inclusion and exclusion criteria separately drawn.
The figures should be presented in the main manuscript instead of supplementary information. No figure legends were provided. There is no table 1 in the files, only graphs without information on the x and y axis are provided.
Unfortunately, a fair review cannot be performed for this manuscript, and therefore needs to be resubmitted again with proper proofreading by the authors.
Author Response
Dear reviewer,
thank you for your extensive review of our submission. Pointing out the method of difference in the abstract the last result sentence has been reworded to „The variance of the difference of the recurrence rates analysis using the surgical date as starting point showed similarities showed similarities in the age subgroup. ” and the result subgroup is clearly named (also in the conclusionn section). As requested the abbreviations have been deleted in the abstract and variable names are used.
As much as we understand (and would like to accompany) our request for a more in depth introduction due to the complexity of the micro-environment, the pathogenesis, tumor genetics and increasing differentiated treatment recommendations this is not possible within the word limit of the journal. Our aim in this submission is to enable future researchers to focus on one tumor subtype/subgroup and present clinical observations, animal studies and laboratory research to expand the knowledge or identify missing evidence. Your simple request for the effect of radiotherapy is worth future studies as to the best of my knowledge thus far the effect of radiotherapy has not been investigated depending on the tumor subgroup. Similar the effect of wound seroma and tumor stimulation. For TNBC this has just had an interesting publication ( PMID: 36325601) but the clinical effect needs to be published. So with the best intentions we had to compromise to accompany the word limits and the vast topic.
The header 1.1 has been deleted and a introduction has been added. “With systemic treatment circulating inactive tumor-cells are increasingly successfully targeted in breast cancer. This is one of the reasons why breast cancer is considered a systemic disease. In the follow up inactive tumor cells may lead to recurrences even after a long time [6-8].”
The method section is structured in paragraphs (general database and data collection, grouping of patients in the study arms, descriptive analysis and Kaplan Meier analysis). A flow chart is included to visualise the study arm differentiation and table 1 provides further details.
A example figure has now been included in the text (figure 2: recurrence pattern of all three study groups) and the Addendum has been extended with subheaders and information on the axis.
We sincerely hope that a fair review is now possible and look forward to your well appreciated constructive comments.
Florian Ebner
Round 2
Reviewer 2 Report
The authors addressed all comments raised by the reviewer. After final proofreading, the revised manuscript meets the standards.
Reviewer 4 Report
The authors have sufficiently addressed the major drawbacks in the revision, therefore, the manuscript can be recommended for publication in its present form to Current Oncology.